# ClassDiffusion: More Aligned Personalization Tuning with Explicit Class Guidance

**Jiannan Huang**[1,2,4]    **Jun Hao Liew**[3]    **Hanshu Yan**[3]    **Yuyang Yin**[1,2]
**Yao Zhao**[1,2]    **Humphrey Shi**[4]    **Yunchao Wei**[*1,2]

[1] Institute of Information Science, Beijing Jiaotong University
[2] Visual Intelligence + X International Joint Laboratory of the Ministry of Education
[3] ByteDance Inc.  [4] SHI Labs@Georgia Tech

## Abstract

Recent text-to-image customization works have proven successful in generating images of given concepts by fine-tuning the diffusion models on a few examples. However, tunning-based methods inherently tend to overfit the concepts, resulting in failure to create the concept under multiple conditions (*e.g.*, headphone is missing when generating "a `<sks>` dog wearing a headphone"). Interestingly, we notice that the base model before fine-tuning exhibits the capability to compose the base concept with other elements (*e.g.*, "a dog wearing a headphone"), implying that the compositional ability only disappears after personalization tuning. We observe a semantic shift in the customized concept after fine-tuning, indicating that the personalized concept is not aligned with the original concept, and further show through theoretical analyses that this semantic shift leads to increased difficulty in sampling the joint conditional probability distribution, resulting in the loss of the compositiona ability. Inspired by this finding, we present **ClassDiffusion**, a technique that leverages a **semantic preservation loss** to explicitly regulate the concept space when learning the new concept. Although simple, this approach effectively prevents semantic drift during the fine-tuning process on the target concepts. Extensive qualitative and quantitative experiments demonstrate that the use of semantic preservation loss effectively improves the compositional abilities of fine-tuning models. Lastly, we also extend our ClassDiffusion to personalized video generation, demonstrating its flexibility.

## 1 Introduction

Thanks to the rapid progress in the diffusion model [31, 48, 55, 59, 63, 65, 67, 68, 72, 92, 95], the field of text-to-image generation has achieved significant progress in recent years. The leading text-to-image models [1, 20, 34, 38, 40, 66, 80] have been successful in generating high-fidelity images that align well with textual inputs. Recently, a significant part of the research [2, 4, 5, 8, 11, 28, 36, 60, 79, 88, 93, 94] has changed their focus from creating high-quality images to improving control over the generated images. Among these works, an important and widely explored research domain is subject-driven personalized generation, which aims to generate new images for a specific concept given some reference images of that concept.

Existing personalization methods [1, 9, 20, 21, 34, 38, 40, 51, 66, 78, 80, 81, 85] can generate images that closely resemble the concept by fine-tuning the base text-to-image model in a specific image set. However, all tuning-based models will inherently suffer from the over-fitting introduced by this process, which leads to weakening in the compositional ability of the model. For example, when generating "a `<sks>` dog wearing a headphone", though the given dog is well reconstructed, the headphone is always missing (Fig. 1). This feature affects the diversity of the generated output in practical use. A commonly accepted explanation within the community [20, 29, 66, 75] attributes this phenomenon to overfitting given a limited number of images. However, the fundamental cause of this overfit remains unexplored. In this work, our aim is to investigate the underlying causes behind the overfitting.

---

*Corresponding author.

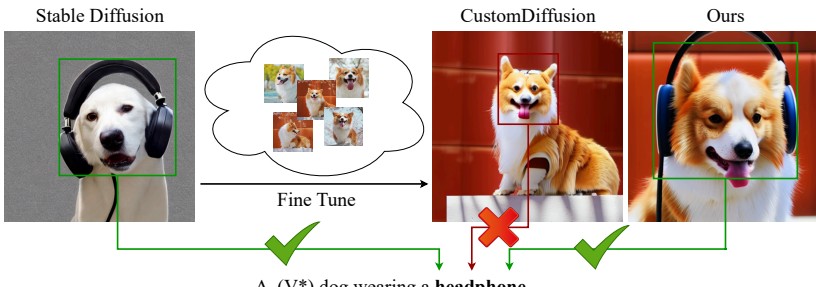

A (V*) dog wearing a **headphone**

Figure 1: The base Stable Diffusion (SD) possesses the capbility to compose the concept of a dog and headphone, generating a dog wearing a headphone. However, we notice that this compositional generation capability is lost during personalization tuning. For example, when using Custom Diffusion (CD) [38], the headphone is missing despite the target corgi is generated successfully. On the other hand, our method can successfully compose the target corgi with the headphone.

Upon initial examination, it appears that the model diminishes some of its original capabilities after personalization tuning. Taking Stable Diffusion (SD) [74] as an example, from Fig. 1, we observe that the base SD model indeed has the ability to combine the concepts of a dog and a headphone. However, after fine-tuning, the model struggles to achieve compositional generation; for instance, while the target concept <sks> (dog) can be generated successfully, the headphone is missing. **We hypothesized that the decline in this compositional ability stems from the semantic drift of the target concept away from its superclass target during fine-tuning.** To better understand this, we conduct some empirical analysis by visualizing the CLIP text-space and cross-attention map activation area in Fig. 3a, 3b. In addition, we also perform theoretical analysis and find that the root cause lies in the semantic bias that reduces the entropy of the probability of the composed conditions, which significantly increases the difficulty to simultaneously sample the target concept combined with other elements.

Based on our experimental findings and theoretical analysis, we introduce ClassDiffusion to address the issue of weakening compositional capacity after fine-tuning. Fig. 2 shows the performance of our method. Our method uses semantic preservation loss to explicitly guide the model to restore the semantic imbalance that arises during the fine-tuning stage. In particular, it narrows the gap between the text embeddings of the target concept and its respective superclass in the textual space. Despite its simplicity, the proposed loss can successfully recover the compositional ability as shown in Fig. 1. Therefore, distinct from prevalent loss design in the community which seek to migrate the overfitting in tuning-based models, our method enhances the model's capacity of following the text prompt while maintaining the concept of a customized subject. Extensive experiments have demonstrated the effectiveness of our method in restoring the compositional generation capability of the base model. Furthermore, we explore the potential of our approach in personalized video synthesis, showcasing its ability in recovering the semantical space of the generative model. In addition, we found that the CLIP-T metric can hardly reflect the actual performance of personalized generation. Therefore, we introduce the BLIP2-T metric, a more equitable and effective evaluation metric for this particular domain. To summarize, the contributions of our work are:

- We offer a thorough examination to understand why existing tuning-based subject-driven personalized methods inherently suffer from the loss of compositional ability. This is elucidated through both experimental observations and theoretical analysis.

- We propose ClassDiffusion, a simple technique to recover the compositional capabilities lost during personalized tuning.

- Extensive experiments demonstrate that the proposed technique achieves improved personalization ability in image and video generation tasks.

## 2 RELATED WORK

**Text-to-Image Generation and Its Control**   Text-to-image generation is designed to generate high-quality, high-fidelity images that are aligned with textual prompts. This field has been under research

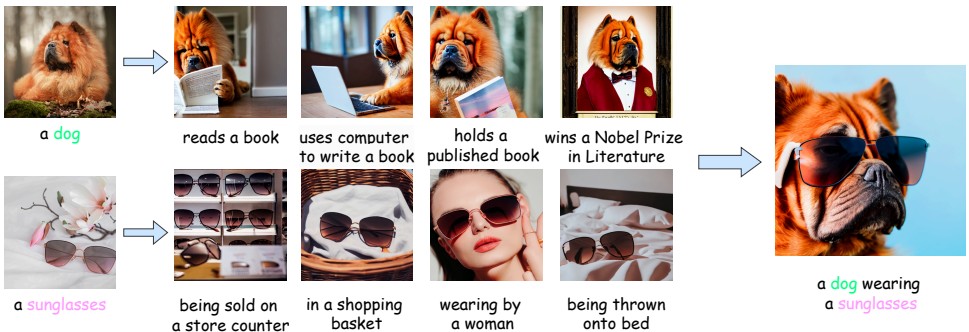

Figure 2: A qualitative result of two small stories produced by our model. The above showcases a bear's literary journey: from reading a book to ultimately earning a Nobel Literature Prize. The below shows the fate of a sunglasses. Finally, the bear gets the sunglasses. It shows a potential real-world application due to our model's high performance.

for an extended period. Recently, the field of Text-to-image generation has made significant progress with extensive research in Generative Adversarial Network (GAN) [23, 24, 53, 98], Variational Autoencoder (VAE) [6, 13, 37, 76], and Diffusion models [31, 48, 55, 59, 63, 65, 67, 68, 72, 95]. The diffusion models achieve a new state-of-the-art (SOTA) in unconditional image generation. Numerous works [12, 33, 43, 46, 55, 61–63, 67, 91, 96, 99] have been done to make the image generated by diffusion models more aligned with the textual prompts. Among them, Stable Diffusion [65] is a widely recognized model in the field, utilizes a cross-attention mechanism to integrate textual conditions into the image generation process and employs the Latent Diffusion Model, which maps the image to latent space [65]. Our research is based on the Stable Diffusion framework due to its adaptability and wide use in the community. Furthermore, different methods exist for controlling generative models. The primary categories for controlling generative models typically include Text-guided [7, 18, 22, 60, 64], Image-guided [1, 20, 34, 38, 40, 66, 80], Additional Sparse conditions [2, 4, 5, 8, 11, 28, 36, 49, 60, 79, 88, 93, 94], Brain-guided [3, 10, 19, 47, 54, 57, 74], Sound-Guided [60, 89], and some universe control [42, 49, 90]. Text-guided control utilizes textual descriptions to directly influence the outcome, guiding the model based on specific verbal instructions. Our method focused on the text-guided controllable generative model.

**Subject-Driven Personalized Generation** Subject-driven personalized generation is focused on creating images based on reference images. Recent works [1, 9, 20, 21, 34, 38, 40, 51, 58, 66, 73, 78, 80, 81, 85, 87] have explored techniques for producing striking resemblance images in multiple ways. One of the primary ways is to fine-tune the base text-to-image models. Furthermore, there has been a significant effort in research [14, 29, 32, 35, 38, 39, 71, 80, 84] aimed at integrating various concepts in personalization. While striking resemblance images are produced, fine-tuning the base model on a small set of images leads to overfitting, resulting in unexpected issues. One prevalent issue discussed in prior research is the decrease in diversity. Recent studies have proposed various methods to address these issues. For instance, DreamBooth [66] introduced the Class-Specific Priori Loss, which effectively addresses diversity reduction by recovering the class's prior knowledge. However, it does not effectively maintain the ability of the model to follow the text prompt. Another common issue is the inability to generate images under multiple conditions. Some Recent Research [29, 30, 40, 50, 75] proposed some methods to migrate this appearance. However, the underlying reasons for this phenomenon have not been thoroughly investigated. Our research endeavors to explore these reasons and develop solutions to overcome this challenge.

## 3 METHOD

### 3.1 PRELIMINARY

**Text-to-Image Diffusion Model** Stable Diffusion [65] is widely used in image generation task. For any input image, Stable Diffusion first transforms it into a latent representation $x$ using the encoder $\varepsilon$ of a variant auto-encoder [37]. For any input image, Stable Diffusion first transforms it into a

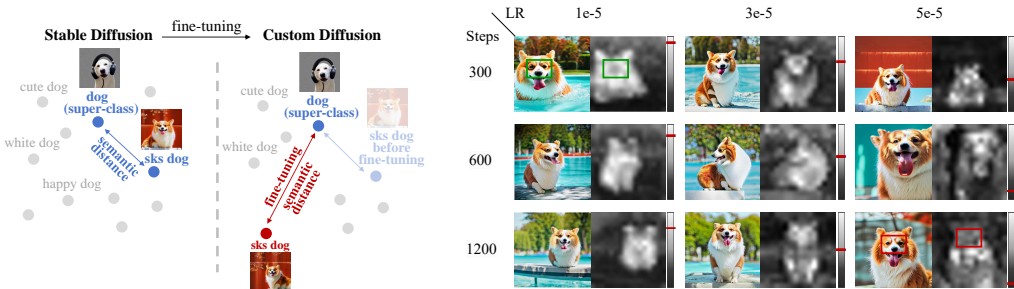

(a) Comparison of distances in CLIP text space.    (b) Visualization of the cross-attention map activation area.

Figure 3: (a) Each dot represents the position of a phrase combining an adjective and "dog" in the CLIP text-space. After fine-tuning, customized concepts move further away from the the distribution of super-class. (b) Visualization results of cross-attention map activation maps corresponding to the dog token. The bar chart on the right shows the average activation level in the dog area. Experiments show that the activation strengths of the corresponding classes decrease with the increase of the learning rate and the total number of training steps. These demonstrate that the customized concepts likely no longer belong to the super-class, resulting in a loss of super-class semantic information, such as wearing a headphone.

latent representation $x$ using the encoder $\varepsilon$ of a variant auto-encoder [37]. The diffusion process then operates on $x$ by incrementally introducing noise, resulting in a fixed-length Markov chain represented as $x_1, x_2, ..., x_T$, where $T$ is the chain's length. Stable Diffusion uses a UNet architecture to learn the reverse of this diffusion process, predicting a denoised version of the latent input $x_t$ at each timestep $t$ from 1 to $T$. In the context of text-to-image generation, the text prompts' conditioning information $y$ is encoded into an intermediate representation $\tau_\theta(y) = c$, where $\tau_\theta$ is a pre-trained CLIP [61] text encoder. The primary objective in training this text-to-image diffusion model involves optimizing this transformation and prediction process, and it can be expressed as:

$$\mathcal{L}_{recon} = \mathbb{E}_{x,y,\epsilon,t}\left[\|\epsilon - \epsilon_\theta(x_t, t, \tau_\theta(y))\|_2^2\right] \tag{1}$$

where $\epsilon$ and $\epsilon_\theta$ represent the noise samples from the standard Gaussian noise $\epsilon \sim \mathcal{N}(0, 1)$ and predicted noise residual, respectively.

**Subject-driven Diffusion Model**    Although text-to-image models have achieved remarkable performance, their controllability is limited. To personalize the generated outputs, DreamBooth[66] fine-tunes the diffusion U-Net to fit several target concept images. Custom Diffusion[38] introduces a new modifier token V* in front of the category name and optimizes only the key and value matrices in the cross-attention layers, thereby improving efficiency.

### 3.2 EXPERIMENTAL ANALYSIS

We begin by observing simple experimental test cases to realize that the loss of compositional ability after personalization tuning is a common phenomenon. We then analyze the underlying logic through visualizations of the CLIP text-space and cross-attention strength map. Finally, we conduct a theoretical analysis to support our hypothesis.

**Simple experimental test cases.** As shown in Fig. 1, we observe a loss of compositional ability after fine-tuning. We then conduct additional test cases and find that the headphone concept is not the only one affected; other concepts also experience a similar loss. Furthermore, this situation occurs in both the dog case and other classes.

**Semantic drift in CLIP text-space.** To elucidate the reasons, we project text into the CLIP text space and use two dimensions for simplified visualization in Fig. 3a. Each dot represents phrases composing the super-class ("dog" in this case) with different adjectives (*e.g.*, "a cute dog", "a white dog" *etc.*). Before fine-tuning, the customized concepts (<sks> dog) have special meaning and are at a similar distance from the super-class as other words. After fine-tuning, we observe a significant increase in the distance of the target concept from its corresponding super-class. This indicates that

the semantics of the personalized concepts change during fine-tuning. In short, the model increasingly fails to recognize that personalized concepts belong to the dog category. This shift may lead to an inability to access the knowledge associated with the super-class(like wearing a headphone). More details can be found in Appendix D, E.

**Reduction of cross-attention activation strength.** We further investigate the model by visualizing the cross-attention layers in Fig. 3b. The attention maps indicate the activation area of super-class words in cross-attention layers. It shows that while the "dog" token activates the relevant region in the image, its activation level is notably lower than that of the pre-trained model. Furthermore, its activation level decreases with the increase in epochs and the learning rate. These findings align with our observations in the CLIP text space and provide support for hypothesis that the customized concepts are increasingly not recognized as part of the super-class during fine-tuning.

Next, we theoretically analyze why the semantic drift of personalized phrases results in the weakening of the compositional ability from the respective reduction in the entropy of composable conditional probability.

## 3.3 THEORETICAL ANALYSIS

Drawing on the insights of [45], a trained diffusion model can be seen as implicitly defining an Energy-Based Model (EBM) [17]. This perspective allows us to build on prior research in composing EBMs and adapting them for use in diffusion models. Building on the work of [16] in the context of generating images with multiple attributes and Bayes' theorem, the conditional probability can be decomposed as:

$$p(x|c_{class}, c_1, c_2, \ldots, c_i) \propto p(x, c_{class}, c_1, c_2, ..., c_3) = p(c_{class}|x)p(x) \prod_{i \in T} p(c_i|x) \tag{2}$$

$$= p(c_{class}|x)p(x) \prod_{i \in T} \frac{p(c_i)p(x|c_i)}{p(x)} \tag{3}$$

where $T$ is all the set of conditions in prompts except for the class, $p(c_i)$ represents the probability of occurrence of condition $c_i$ in the training dataset and can be regarded as a constant for large-scale pre-training models. $p(c_i|x)$ represents an implicit classifier, denoting the probability of categorizing a concept as $c_{class}$. Specifically, $p(c_{class}|x)$ represents a specific implicit classifier for the super-category. Thus, we have:

$$p(x|c_{class}, c_1, c_2, \ldots, c_i) \propto p(c_{class}|x)p(x) \prod_{i \in T} \frac{p(c_i)p(x|c_i)}{p(x)} \tag{4}$$

Denoted $p(x) \prod_{i \in T} \frac{p(c_i)p(x|c_i)}{p(x)}$ as $d(x)$, $p(c_{class}|x)$ as $q(x)$, and $p(x|c_1, c_2, \ldots, c_i)$ as $a(x)$. The entropy of $a$ is calculated as:

$$\mathrm{H}(a) = -\sum_x q(x)d(x)[\log(q(x)) + \log(d(x))] \tag{5}$$

After fine-tuning, the components of $d(x)$ change only slightly and can be treated as unchanged, and the implicit classifier $p_\theta(c_{class}|x)$ changes to $p_{\theta'}(c_{class}|x)$, Thus the difference in entropy before and after can be expressed as:

$$\triangle \mathrm{H} = \sum_x q_\theta(x)d(x) \left[\log(q_\theta(x)) + \log(d(x))\right] \tag{6}$$

$$- \sum_x q_{\theta'}(x)d(x) \left[\log(q_{\theta'}(x)) + \log(d(x))\right]$$

$$= d(x) \sum_x \{[q_\theta(x) \log q_\theta(x) - q_{\theta'}(x) \log q_{\theta'}(x)] \tag{7}$$

$$+ \log d(x) \left[q_\theta(x) - q_{\theta'}(x)\right]\}$$

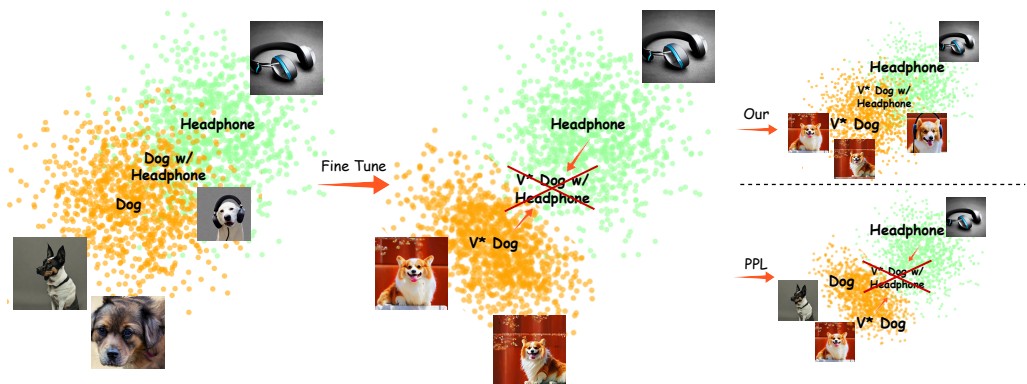

Figure 4: The orange and green point sets represent the distributions of dogs and headphones, respectively, and their overlapping regions represent their joint probability distributions. During the tuning process, the conditional distribution of dogs and headphones shrinks, which gradually increases the difficulty of sampling. Unlike the Prior Preservation Loss (PPL) in DreamBooth [66], which aims to maintain class diversity, our proposed Semantic Preservation Loss (SPL) focuses on recovering the semantic space of the customized concept. This approach enables our method to synthesize images that are more consistent with the text prompt.

Based on our observations in Fig. 3a, 3b we can show that $q_\theta(x) > q_{\theta'}(x)$, combining the properties of probability theory and the monotonically decreasing nature of $x \log x$ at $(0,1)$, we have:

$$q_\theta(x) \log q_\theta(x) - q_{\theta'}(x) \log q_{\theta'}(x) < 0; \log d(x) < 0 \qquad (8)$$

Thus, we have:

$$\triangle \mathrm{H}(a) < 0 \qquad (9)$$

As a result, it is more difficult to sample from our demanded conditional distributions under $c_{\text{class}}, c_1, ..., c_i$ conditions than before the fine-tuning, leading to the phenomenon that the combining ability is weakened after the fine-tuning. We will discuss the theoretical reasons here in more detail in Appendix B. Fig. 4 illustrates the changes in the distribution during this process that lead to a weakening of the compositional generation capability.

The diminished combinatorial capacity is due to the increased difficulty in sampling from joint conditional probabilities caused by shifts in the semantics of customisation concepts. Therefore, in order to reduce the difficulty of sampling in joint conditional probability distributions, we need to recover the original semantic space of the text, i.e. to recover the semantic distance between custom concepts and superclasses in the semantic space that has been distanced by the fine-tuning process. So, in the next section, we present our proposed semantic preservation loss to mitigate the semantic drift that occurred during fine-tuning.

## 3.4 SEMANTIC PRESERVATION LOSS

As analyzed above, the key challenge lies in preserving semantic information during fine-tuning to reduce the difficulty of sampling from the joint conditional distribution. To address this, we propose a novel loss function aimed at constraining semantic variation throughout the fine-tuning process.

Specifically, considering that there are $N$ special tokens, each representing a customized concept. During the process, the embeddings associated with these tokens are fine-tuned to align with the target concepts. Our loss function is designed to minimize the semantic distance between the phrase containing the special token (*e.g.*, a photo of a V* dog) and the phrase containing only the class word (*e.g.*, a photo of a dog). By labeling the training prompt as $P_{tp}$ (*e.g.*, a photo of a V* dog), and the class prompt as $P_{cp}$ (*e.g.*, a photo of a dog), we use a text encoder to get their embeddings $E_{tp}$ and $E_{cp}$ in Stable Diffusion's semantic space. The semantic preservation loss (SPL) is calculated by the sum of the cosine distance of their text embeddings. Formally speaking, our proposed SPL can be

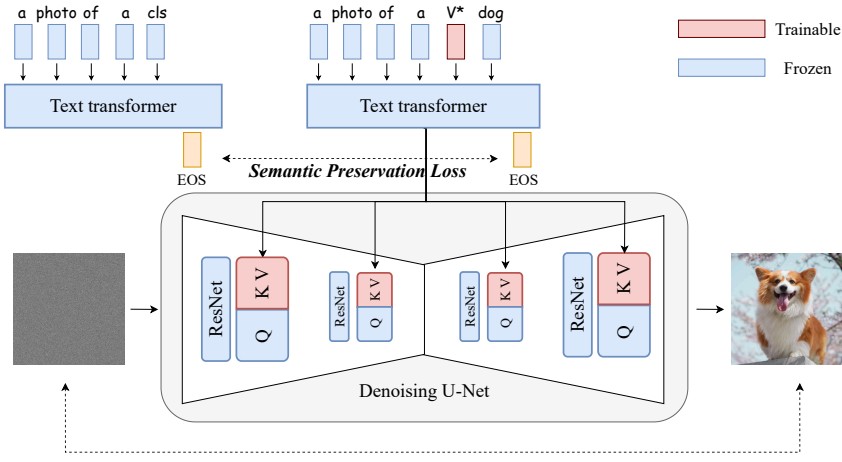

Figure 5: The framework of ClassDiffusion. The personalization fine-tuning strategy is based on Custom Diffusion [39], which primarily fine-tunes the K and V parameters in the transformer block. Our **semantic preservation loss (SPL)** is calculated by measuring the cosine distance between text features extracted from the same text transformer (using EOS tokens as text features following CLIP) for phrases with personalized tokens and phrases with only super-class.

expressed by the following equation:

$$\mathcal{L}_{sp} = \sum^{N} \sum^{B} \sum^{L} D_c \left( E_{SC}, E_C \right) \tag{10}$$

where $B$ represents the batch size, $L$ denotes the hidden dimensions of the text encoder, and $D_c$ implies the cosine distance. We can represent the final training objective as:

$$\mathcal{L} = \mathcal{L}_{recon} + \lambda \mathcal{L}_{sp} \tag{11}$$

The overview of our proposed model is shown in Fig. 5

## 4 EXPERIMENTS

### 4.1 EXPERIMENT DETAILS

**Implementation details** Our method is built on Stable Diffusion V1.5, with a learning rate $10^{-6}$, and batch size 2 for fine-tuning. We used 500 optimization steps for a single concept and 800 for multiple concepts, respectively. During inference, the guidance scale is set to 6.0 and the inference steps are set to 100. The semantical preservation loss weight is set to 1.0 during all experiments. All experiments are conducted on 2×RTX4090 GPUs. Our method uses $\sim$ 6 min for the generation of single concepts and $\sim$ 11 min for the generation of multiple concepts.

To better preserve the semantic space, we compute SPL between text embeddings embedded in the semantic space of the Stable Diffusion model. Therefore, we utilize the CLIP [61] text encoder from Stable Diffusion v1.5 [63], specifically clip-vit-large-patch14 [47], to extract the text embeddings of phrases. Following common practice, we use the End of Sequence (EOS) token to represent the semantics of embeddings.

**Baselines** We compare our method with state-of-the-art (SOTA) competitors, including Dream-Booth [66], Textual Inversion [20], Custom Diffusion [38], NeTI [1], SVDiff [29]. For DreamBooth, CustomDiffusion, and Textual Inversion, we used the diffusers [77] version of the implementation. For NeTI, we use its official implementation. Given that SVDiff does not have an official open-source repository. For SVDiff, we use the implementation of [69]. All training parameters follow the recommendations of the official paper. To ensure fairness of comparison, all these baselines are built on Stable Diffusion V1.5.

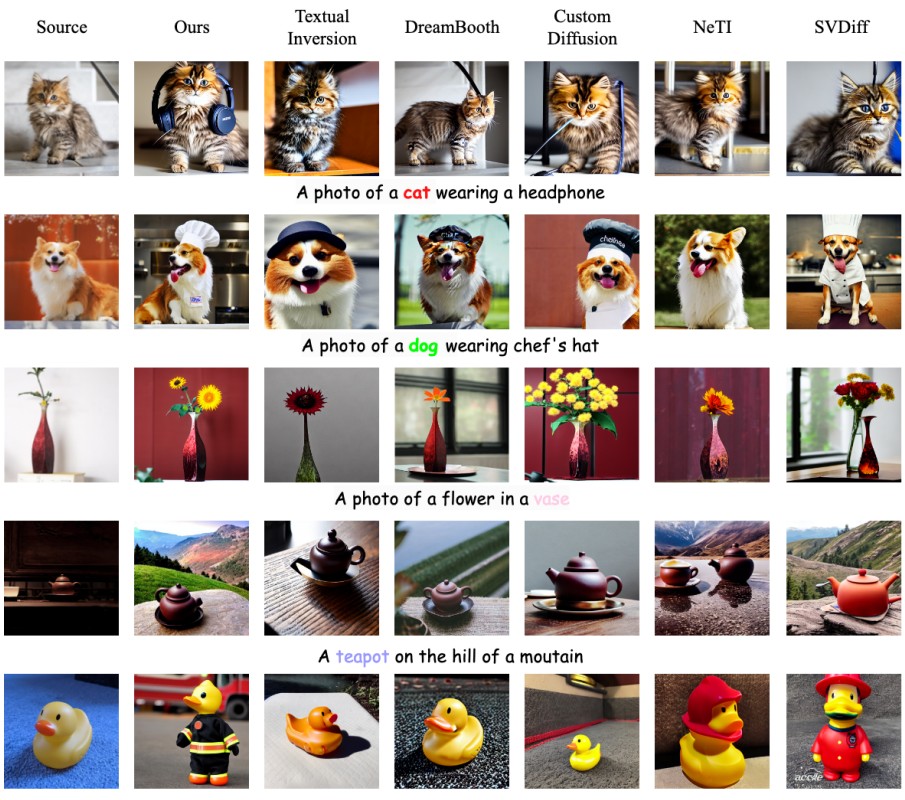

Figure 6: Qualitative comparison between our method and baselines with single given concept. Our method generates images that align with the prompts, surpassing all baselines.

**Datasets** Following previous work [29, 66, 75], we conduct quantitative experiments on DreamBooth Dataset [66]. It contains 30 objects including both live objects and non-live objects. In addition, we used images from the Textual Inversion Dataset [20] and CustomConcept101 [38] in qualitative experiments.

**Evaluation metrics** We assess our approach using three metrics: CLIP-I, CLIP-T, and DINO-I. CLIP-I calculates the visual similarity between the produced images and the target concept images by utilizing CLIP [61] visual features. CLIP-T evaluates the similarity between text prompts and images. If one baseline contains the special token S*, it will be replaced with a prior class word. In the case of DINO-I, we evaluate the cosine similarity between the ViT-S/16 DINO [56] embeddings of the generated images and the concept images. Further, we note the impact of CLIP's outdated performance on the fairness of the evaluation. Therefore, we introduce the BLIP2-T Score, which calculates the similarity between text features extracted from BLIP2's Q-former and image features extracted from Vision Encoder as a score. This metric is designed by calculating the similarity between image and text embeddings extracted by the BLIP2 model. Our approach involved utilizing the Transformer [82] implementation and the fine-tuned weights of BLIP2-IMT on CoCo [44], with ViT/L [15]. This new metric aims to offer a more equitable and efficient evaluation measure for future studies in this field. Empirical findings from various studies [25, 40, 41, 70, 83, 86] indicate that BLIP2 outperforms CLIP significantly in the assessment of text-image alignment.

## 4.2 QUALITATIVE & QUANTITATIVE EXPERIMENTS

**Qualitative Experiments** We compare our method with DreamBooth [66], Textual Inversion [20], NeTI [1], SVDiff [29], and Custom Diffusion [38] on challenging prompts. The results depicted in Fig. 6 demonstrate the outcomes obtained from these prompts. Fig. 2 shows a story of a given dog and sunglasses. The experimental findings indicate a substantial superiority of our approach over other techniques regarding alignment with text prompts, without any decline in similarity to the specified concept. More qualitative results are shown in Appendix. J. Also, we conduct

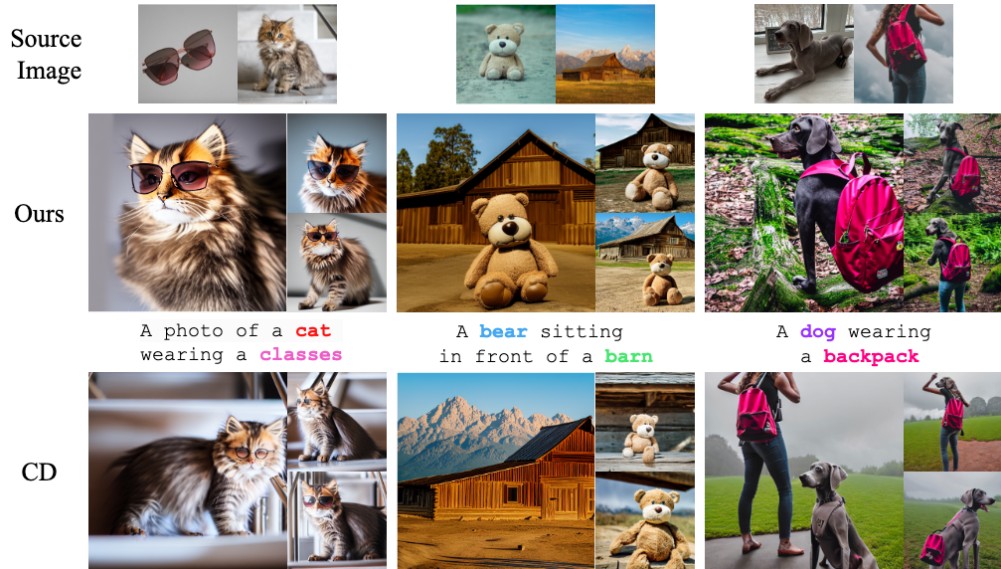

Figure 7: Qualitative comparison between our method and Custom Diffusion(CD) in multiple concepts. Our method has better text alignment than custom diffusion.

qualitative experiments with multiple concepts on the combinations `<cat>`, `<sunglasses>`; `<bear>`,`<barn>`; `<dog>`,`<backpack>`; Fig. 7 shows the results of the experiments. The experiments show that our method can be aligned with prompts better than custom diffusion in multi-concept generation.

**Quantitative Experiments** Following the previous work, we used 20 concepts for quantitative experiments. For Single Concept text similarity metrics (CLIP-T, BLIP2-T), we followed the 25 prompts used in [66], sampling 20 images per prompt. The results of the experiment are shown in Tab. 1. Experimental results show that our method obtains new SOTA on each text similarity metric, indicating that we have good compositional generation capability.

Table 1: Quantitative Results on all Metrics and Results of the User Study. The last two columns display the win rates of our method compared to other approaches in the user study, evaluated in terms of text similarity and image similarity. A win rate exceeding 50% (highlight by ✓) indicates that our method outperforms the compared methods on the corresponding metric, as judged from a human perspective.

| | Method | CLIP-T↑ | CLIP-I↑ | DINO-I↑ | BLIP2-T↑ | TIFA↑ | User-T Win Rate ↑ | User-I Win Rate ↑ |
|---|---|---|---|---|---|---|---|---|
| **Single Concept** | DreamBooth [66] | 0.249 | **0.855** | **0.700** | 0.295 | 0.559 | 95.4% ✓ | 42.1% |
| | Textual Inversion [20] | 0.242 | 0.825 | 0.631 | 0.308 | 0.505 | 95.1% ✓ | 75.0% ✓ |
| | Custom Diffusion [38] | 0.286 | 0.837 | 0.693 | 0.416 | 0.746 | 79.1% ✓ | 40.0% |
| | NeTI [1] | 0.290 | 0.838 | 0.648 | 0.329 | 0.607 | 78.8% ✓ | 70.0% ✓ |
| | SVDiff [29] | 0.293 | 0.834 | 0.606 | 0.418 | 0.835 | 56.6% ✓ | 95.8% ✓ |
| | Our | **0.300** | 0.828 | 0.673 | **0.460** | **0.843** | - | - |
| **Multiple Concepts** | Custom Diffusion [38] | 0.282 | 0.813 | **0.636** | 0.380 | - | - | - |
| | Our | **0.320** | **0.821** | 0.604 | **0.477** | - | - | - |

## 4.3 ADDITIONAL EXPERIMENTS

**User Study** We also performed user study to validate the effectiveness of our method. We used the same set of images generated in the Section 4.2 for user study, details of which are available in Appendix. I. The results of the user study are located in Tab. 1. The numbers in the table indicate at what percentage our method is considered by humans to be superior to the compared methods. The result of the user study shows that our method outperforms all methods in text similarity (> 50%).

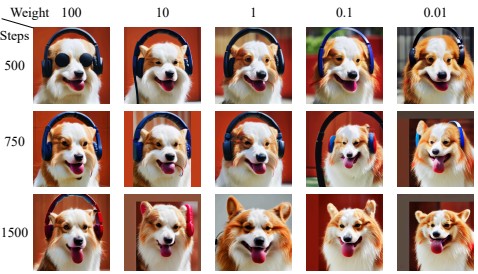

Figure 8: Generation results for the prompt "a photo of a dog wearing a headphone" with different step counts and SPL weights. All results are generated using the same random seed.

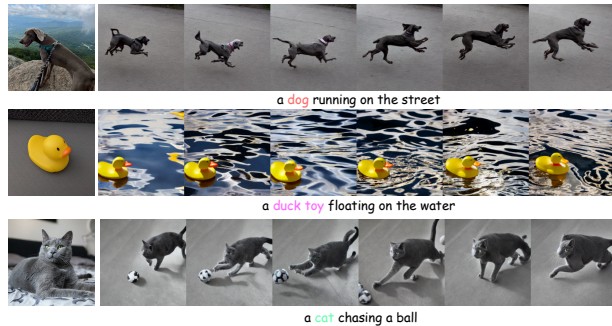

Figure 9: Result of generated videos, showing good textual alignment and similarity of given concepts.

Although our method does not outperform all methods in image similarity, given the high CLIP-I scores, our method still produces images that are highly consistent with the given concept.

**Personalized Video Generation** We investigate the implementation of our method in personalized video generation. We utilized AnimateDiff V2 [26, 27] for video generation, configuring parameters to a resolution of $512 \times 512$, a guidance scale of 7.5, and 25 inference steps. The outcomes of the video generation process are illustrated in Fig. 9. Utilizing AnimateDiff, our technique produces videos that exhibit strong textual and conceptual coherence without the need for additional training. This demonstrates that our approach, which aligns personalized phrases with superclass-centric semantics, can generate engaging videos with dynamic generation capabilities stemming from pre-training, along with the ability to transition across corresponding domains.

**Ablation Experiments** We studied the influence of different weights of semantic preservation loss (SPL). The results show that higher SPL weights preserve combining ability better. At s SPL weight of 100, all steps successfully depict "wearing a headphone." However, at lower weights of 0.1 and 0.01, the headphone details diminish by 750 steps and disappear by 1500 steps. On the other hand, lower SPL weights restore specific concept features more effectively.

## 5 CONCLUSIONS

In this work, we highlight the problem of weakened compositional ability due to individualized fine-tuning and provide an analysis of the causes of this problem from experimental observations and information-theoretic perspectives. We discovered that this weakening effect is primarily attributed to the semantic shift of the customized concepts throughout the fine-tuning process. As the model undergoes fine-tuning, the representations of these concepts gradually drift away from their original meanings, leading to a misalignment with the intended semantics. This semantic drift complicates the model's ability to accurately sample from the joint conditional distribution, ultimately hindering its performance in generating or understanding the intended outcomes based on the fine-tuned concepts. We then introduce a new approach, termed ClassDiffusion, which mitigates the weakening of compositional ability by restoring the original semantic space. Finally, we present comprehensive experimental results showcasing the efficacy of ClassDiffusion and the fresh perspectives it offers on interconnected fields.

## 6 ACKNOWLEDGEMENT

This work was supported in part by the National Key R&D Program of China under Grant No. 2022YFC3310200, the National Natural Science Foundation of China (No.92470203, U23A20314), Beijing Natural Science Foundation (No. L242022 ), and the Fundamental Research Funds for the Central Universities (No. 2024XKRC082)

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

## A    VISUALIZATION AFTER FINE TUNNING

In section 3, we present visualization results for the text space and the cross-attention layer of other methods, highlighting the semantic bias that emerges in the text space, leading to a decline in compositional ability. To further affirm the effectiveness of our method and our hypothesis, we also visualize the textual feature space and the cross-attention layer with our method in this section. The visualization results are depicted in Fig. 10. The ranking of the distance is 36 out of 71, as opposed to 26 out of 71 before fine-tuning and 67 out of 71 for other methods. A comparison with the visualizations in Fig. 3b and Fig. 3a reveals that our model effectively addresses the semantic drift in the text space.

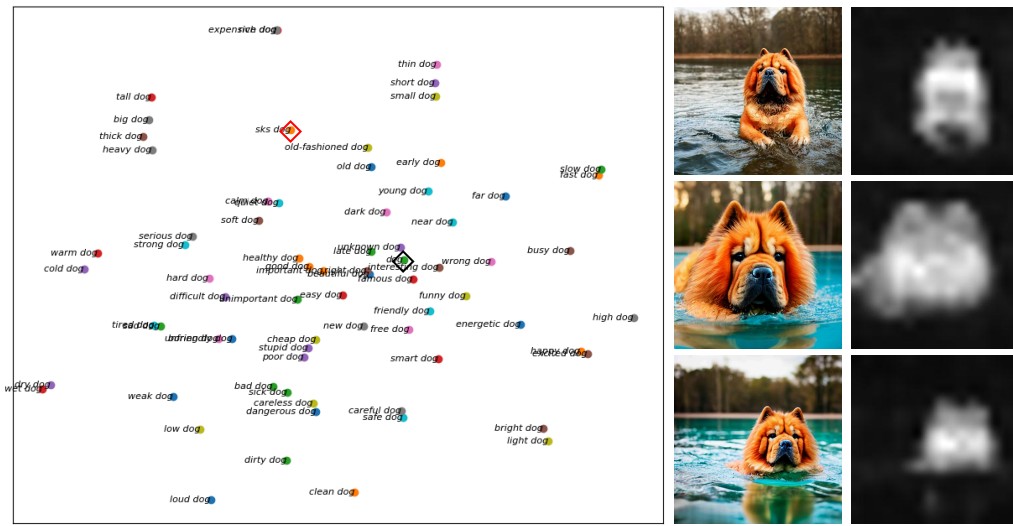

Figure 10: Visualization results after fine-tuning of our approach.

## B    ENTROPY REDUCTION DURING THE FINE-TUNING

In section 3.3, we provide a solid theoretical analysis for the weakening of compositional ability due to the semantic drift from the perspective of information theory and probability distributions. In this section, we will discuss it in a more detailed way.

In the field of subject-driven personalization generation, two manifest phenomena are caused by overfitting: weakening of diversity in classes of given concepts and weakening of compositional ability. In addition to the calculations mentioned in the main body of the text, the entropy of combined conditional probability can also be calculated as conditional entropy:

$$\mathrm{H}(X|c_1, c_2, \ldots, c_i) \tag{1}$$

Make the given concept into a series of specific conditions: $c_{s1}, \ldots, c_{si}$, each condition describes one of the features of the given concept. The entropy after fine-tuning will be:

$$\mathrm{H}(X|c_1, c_2, \ldots, c_n, c_{s1}, \ldots, c_{si}) = \mathrm{H}(X|c_1 \ldots, c_n) - I(X|c_1 \ldots, c_n; c_{s1}, \ldots, c_{si}) \tag{2}$$

Where $I$ represent the mutual information, According to [52], we have:

$$I(X|c_1 \ldots, c_n; c_{s1}, \ldots, c_{si}) \geq 0 \tag{3}$$

$$\mathrm{H}(X|c_1, c_2, \ldots, c_n, c_{s1}, \ldots, c_{si}) < \mathrm{H}(X|c_1, c_2, \ldots, c_i) \tag{4}$$

Coarse & Poor Performance

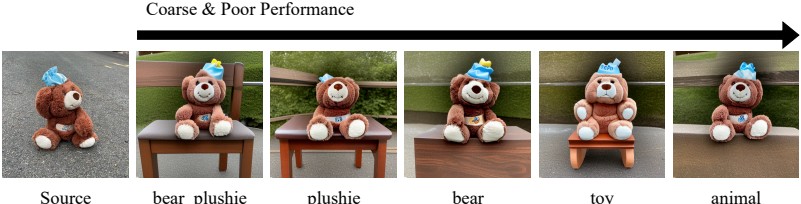

Source     bear_plushie     plushie     bear     toy     animal

Figure 11: Visualization of generating "a  sitting on the chair.". This  is used in both prompt and class token. Experiments show that a fine-grained center word will benefit our proposed method.

However, the entropy reduction here is different from the entropy reduction in the main text. The entropy reduction here leads to a reduction in the diversity of the generated images. Specifically, when the cue word is "a photo of a dog", the image generated is closer to the given concept than to the diversity of dogs.

## C  FINE-GRAINED EXPERIMENTS

At the core of our approach, we want the semantics of personalized phrases to be closer to the category-centered words. In this section, we explore the effect of different category-centered words on the results for the same given concept. Fig. 11 shows the training results using different category-centered words. The results show that the use of different center words leads to significant differences in performance, and a fine-grained center word benefits our method. Also, it indicates the importance of recovering the semantical space of the customized concept.

## D  PROMPT USED IN THE VISUALIZATION OF CLIP TEXT SAMPLE SPACE

In this section, we provide a realistic visualization of the schematic in Fig. 3a, and discuss the prompt to generate the 70 phrases that include adjectives and super-categories "dog", and the whole 71 adjectives.

The realistic visualization of the schematic is:

The prompt we use is:

*Please help me generate some adjectives that can describe an attribute of a dog in a photo.*

The adjectives we use are:

| | | | | |
|---|---|---|---|---|
| *beautiful* | *happy* | *sad* | *tall* | *short* |
| *bright* | *dark* | *big* | *small* | *young* |
| *old* | *fast* | *slow* | *warm* | *cold* |
| *soft* | *hard* | *heavy* | *light* | *strong* |
| *weak* | *good* | *bad* | *rich* | *poor* |
| *thick* | *thin* | *expensive* | *cheap* | *quiet* |
| *loud* | *clean* | *dirty* | *smart* | *stupid* |
| *interesting* | *boring* | *new* | *old-fashioned* | *safe* |
| *dangerous* | *healthy* | *sick* | *easy* | *difficult* |
| *right* | *wrong* | *high* | *low* | *near* |
| *far* | *early* | *late* | *wet* | *dry* |
| *busy* | *free* | *careful* | *careless* | *friendly* |
| *unfriendly* | *important* | *unimportant* | *famous* | *unknown* |
| *excited* | *calm* | *serious* | *funny* | *tired* |
| *energetic* | | | | |

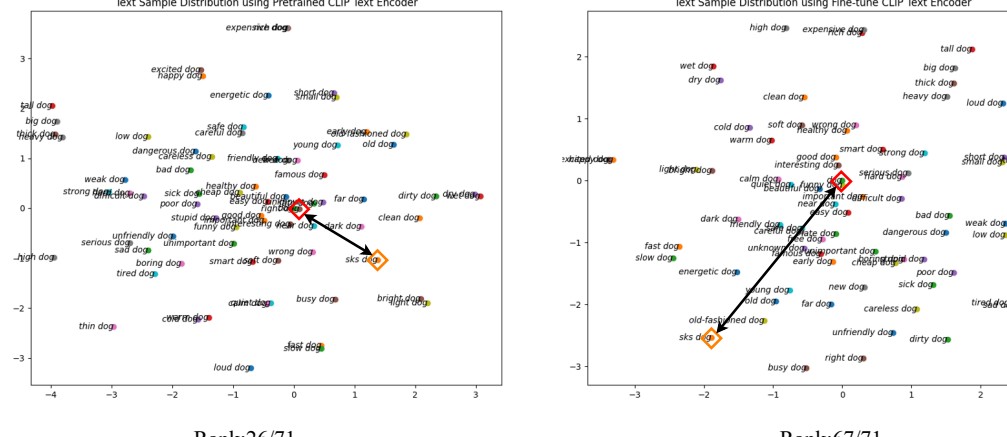

Rank:26/71                                    Rank:67/71

Figure 12: Visualization of the CLIP sample space. Using ChatGPT, we created 70 phrases containing adjectives related to the superclass of dogs. Subsequently, text features derived from these phrases were processed through the CLIP text encoder, downscaled, and their distance from the central point of the superclass (representing a dog image) was calculated. The comparison between the pre-trained model (illustrated in the left figure) and the fine-tuned model (depicted in the right figure) indicates that in the pre-trained model, phrases with special tokens ranked 26 out of 71, while in the fine-tuned model, they ranked 67 out of 71. Moreover, it is evident that phrases containing special tokens are situated further away from the central point of the superclass.

## E   2D TEXT SPACE DISTANCE CALCULATION

In this section, we discuss how we visualized CLIP's text sample space in Fig. 3a. First, we collect 70 phrases that combine adjectives with words representing the class of given concepts (e.g., "a happy dog" or "a cool dog"). Using the CLIP text encoder, we extract text embeddings for these phrases. To visualize the semantic space and intuitively track modifications within it, we use t-SNE to reduce these high-dimensional embedding vectors to 2D. An overview of this result are shown in Fig.3(a), meanwhile we show the real experimental results in Fig.12. Formally, we use the adjectives generated which are described in Section. D as the initialize set $S$, and use the following pseudocode to get a 2D point set $T$:

---

**Algorithm 1** Algorithm to Convert Character Set to 2D Point Set

---

1: **Input:** Initial character set $S$
2: **Output:** 2D point set $T$, Distance set $Dis$
3: $E \leftarrow$ CLIP text encoder encoding$(S)$ ▷ Encode the character set to an encoding set
4: $T \leftarrow$ TSNE$(E)$ ▷ Dimensionality reduction of the encoding set to a 2D point set
5: $Dis \leftarrow \{\|T_i - T_{class}\| \mid i = 1, 2, \ldots, |T|\}$ ▷ Calculate the 2D distance to $T_{class}$ for each point in $T$
6: **return** $T, Dis$

---

## F   MULTI-CONCEPTS EXPERIMENTS

In this section, we demonstrate the ability of ClassDiffusion to generate multiple concepts, specifically 3 concepts in one model. Fig 13 shows the result of this experiment. The experiments demonstrate that ClassDiffusion generates high-quality results when combining multiple concepts, validating the effectiveness of our proposed methods.

Figure 13: Qualitative result of generating three concepts.

## G  MORE BASELINE COMPARISON

To further evaluate the performance of our proposed method, we further introduce a new baseline Prospect [97] which aims at solving similar problems we observe. The result of the experiment is shown in Tab. 2. The quantitative results below show that our model achieves superior performance.

| Models | CLIP-T↑ | CLIP-I↑ | DINO-I↑ |
|---|---|---|---|
| Prospect | 0.294 | 0.815 | 0.588 |
| ClassDiffusion(Ours) | **0.300** | **0.828** | **0.673** |

Table 2: Quantitative results comparing with Prospect

## H  QUANTITATIVE ABLATION OF SPL WEIGHT

In this section, we conduct a quantitative ablation experiment on the choose of SPL weight. Tab. 3 shows the result of the experiment result. This result shows that CLIP-T becomes higher (increase in the ability to follow prompts) and DINO-I decreases(decrease in the ability to customize concepts) with increasing SPL weight, which is consistent with our expectations. Meanwhile, we find that SPL is a loss function that is insensitive to weight. Combined with the qualitative observation in Fig. 8, we prefer to choose 1 as the SPL weight.

## I  DETAILS OF USER STUDY

We offer users a comprehensive user study guide that includes user selection criteria which is shown in Fig. 14. Additionally, to maintain fairness, we positioned our method alongside the baseline method randomly to prevent users from showing bias towards either method. Our percentages in Tab. 1 are obtained by calculating the number that chose our model better as a percentage of the overall number that made a preference.

# User Study

The website of User Study is: xxxx-xxxx.com

After entering the correct token, you will see an interaction similar to the following:

**a bowl in the jungle**

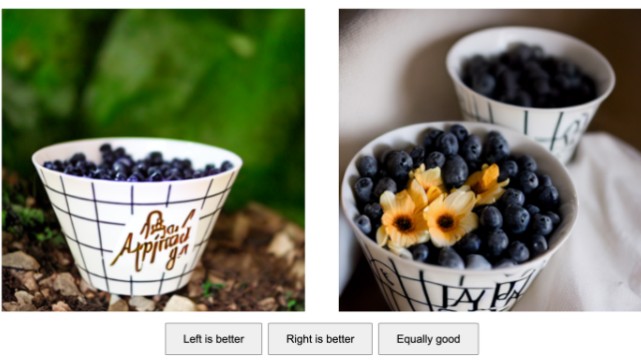

The prompt of images is shown on the top, you should make decision by:

1. If one of the image is aligned with the prompt and another isn't, choose the align one.
2. If none or both of the images is aligned with the prompt, choose the one that is more aligned with the prompt.
3. If you can't make a choice (i.e. you think both diagrams are equally good/bad for the prompt) choose Equally good.

For the example image, you should choose "left is better". This is because although both the left and right images have bowls, the left image has a jungle and the right does not.

Or you will see an interaction similar to the following:

**Reference Image**

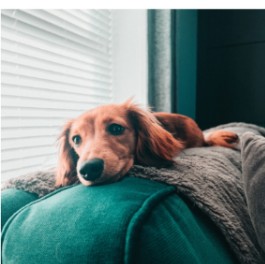

**Generate Image**

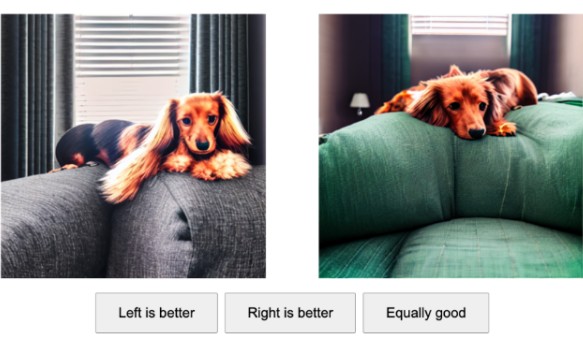

You should choose the one that are more similar to the reference image or choose Equally good if you can not make a decision.

Figure 14: User Study Guide, which describes the user selection criteria and provides an example for reference.

| SPL weight | CLIP-T ↑ | DINO-I ↑ |
|:----------:|:--------:|:--------:|
| 0.01 | 0.299 | 0.677 |
| 0.1 | 0.300 | 0.674 |
| 1 | 0.300 | 0.673 |
| 10 | 0.300 | 0.665 |
| 100 | 0.301 | 0.661 |

Table 3: Performance metrics for different SPL weights.

## J  MORE QUALITATIVE RESULT

In Fig. 6, one generated image is provided for each prompt. Fig. 15 presents more images generated from the same prompts, thereby reinforcing the efficacy of our approach.

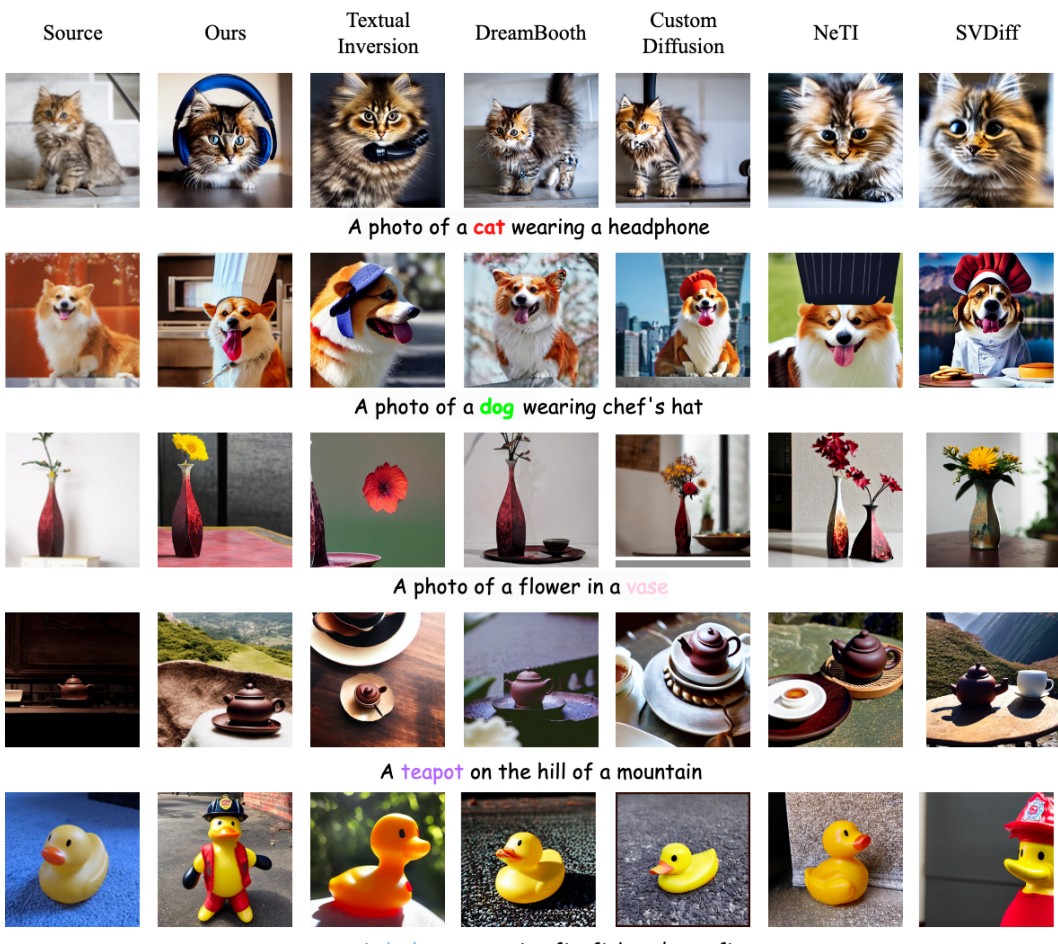

Figure 15: Qualitative result of same prompts the main text.

## K  LIMITATION

In this section, we discuss the limitations of our work. Our work are able to generate images that are aligned with the given prompt while keeping the features of the given concept. However, there are two major limitations in our work:

- Considering that reconstruction of the human face is fine-grained, and the phrase "a photo of a human" or "a photo of a human face" can not include extensive information about humans. Whether our work can transfer to human-driven personalized generation remains explored.

- For objects that have a combination of categories, choosing an appropriate center word requires some experimentation.

## L    SOCIAL IMPACT

The advancements in text-to-image customization through fine-tuning diffusion models, as evidenced by our work on ClassDiffusion, have significant social implications. By enhancing the compositional capabilities of these models, our approach can contribute to a variety of fields, including digital content creation, and education. In the realm of digital content creation, ClassDiffusion enables artists, designers, and marketers to generate more precise and complex images based on textual descriptions. This improvement reduces the time and effort required to produce customized visual content, fostering creativity and innovation. It also allows for the seamless incorporation of personalized elements into digital artworks, advertising materials, and user-generated content, thereby enhancing user engagement and satisfaction. ClassDiffusion can also be a powerful tool in educational settings. Educators can use this technology to create illustrative materials that are tailored to specific learning objectives. For instance, teachers could generate images that accurately depict historical events, scientific concepts, or literary scenes, making learning more interactive and engaging for students. Furthermore, this technology can aid in the development of educational content for diverse learning needs, including materials for students with disabilities.

While the advancements in text-to-image generation hold promise, it is essential to address the ethical considerations associated with their use. Ensuring that these models are free from biases and do not perpetuate harmful stereotypes is crucial. Our work on ClassDiffusion includes measures to mitigate semantic drift, which helps maintain the integrity and accuracy of generated content. Continuous evaluation and updates are necessary to uphold these standards and ensure the technology benefits society as a whole.

