# OpenReview forum: "ClassDiffusion: More Aligned Personalization Tuning with Explicit Class Guidance"
_ICLR.cc/2025/Conference — ICLR 2025 Poster_

### Official Review · Reviewer_BPUN · 2024-10-31

**Soundness:** 3
**Presentation:** 3
**Contribution:** 2
**Rating:** 5
**Confidence:** 3

**Summary:**

For text-to-image customization task, the authors suggest that all tuning-based models will inherently suffer from the over-fitting introduced by this process, which leads to weakening in the compositional ability of the model. They hypothesize that the decline in this compositional ability stems from the semantic drift of the target concept away from superclass target during fine-tuning. They analyze the experimental and theoretical aspects in turn, and introduce ClassDiffusion to address the issue of weakening compositional capacity after fine-tuning.

**Strengths:**

1. The authors explain what they work do with a vivid example, which is easy for readers to understand.
2. The authors theoretically analyze the sampling difficulty based on the joint conditional probability distribution.
3. The paper is written logically and clearly, from experimental phenomenon, theoretical proof, and solution method respectively.
4. The authors extend the proposed method to the video field.

**Weaknesses:**

1. The paper mentions that “CLIP-T metric can hardly reflect the actual performance of personalized generation” but CLIP is still used to calculate Loss (Equation 10).
2. The paper mentions “higher SPL weights preserve combining ability better” and “lower SPL weights restore specific concept features more effectively”, why is the weight of SPL set to 1 in dataset A. Is this the optimal value? It needs to be further discussed.
3. The writing is somewhat redundant. For example, it is not clear how significant Figure 2 is.

**Questions:**

1. The paper mentions that "CLIP-T metric can hardly reflect the actual performance of personalized generation" but CLIP is still used to calculate Loss (Equation 10). The rationale for using CLIP for loss calculation, despite its limitations for evaluation, should be explained.

2. The paper mentions "higher SPL weights preserve combining ability better" and "lower SPL weights restore specific concept features more effectively", why is the weight of SPL set to 1 in dataset A. Is this the optimal value?  An ablation study or justification should be provided for the choice of 1 as the SPL weight, and how they balanced the trade-off between preserving combining ability and restoring specific concept features should be discussed.

3. The writing is somewhat redundant. For example, it is not clear how significant Figure 2 is. The Figure 1 is already shows the performance of proposed method.

4. The paper mentions that "we found that the CLIP-T metric can hardly reflect the actual performance of personalized generation", but does not clearly explain how this is found. Whether it is based on references to other works is very confusing. Evidence or references should be provided to support their claim about the limitations of the CLIP-T metric for evaluating personalized generation. This would strengthen the justification for introducing a new metric.

5. The paper mentions "we introduce the BLIP2-T metric", is this metric proposed by the authors? It should be clarified whether this is a novel metric proposed by the authors or whether it's based on existing work. If it's novel, more details on its development and validation are requested.

6. Is Figure 3a an experimental result or a schematic diagram? If it is a schematic diagram rigorous notation is recommended.  More detailed captions or annotations should be provided to improve clarity.

---

> ### Author Response · Authors · 2024-11-25
>
> **Q1&W1**: The rationale of using CLIP to calculate Loss
>
> **A1**: Thank you for your comment. Actually, I lean towards that they are two isolated problems.
>
> CLIP-T is a metric that is used to evaluate the similarity between **image-text pairs**. Prior studies [1, 2] have demonstrated that CLIP models encounter challenges in distinguishing fine-grained differences in images, primarily due to the nature of their pre-training strategies. Consequently, some mismatched image-text pairs (e.g. prompt is a dog with a smile while the image is a dog without it) may still receive high CLIP-T scores.
>
> However, our approach, SPL, leverages CLIP to calculate the similarity between **two text prompts**. Since SPL operates within the same modality (text), it remains unaffected by the cross-modal gap problem encountered in CLIP-T. Besides, in order to get the embeddings embedded in the Stable Diffusion(SD)'s semantic space, which is beneficial for preserving the original semantic space, we use the text encoder of SD, specifically CLIP, to extract text embeddings. We will update it in our latest version of the manuscript.
>
> [1] Tong, Shengbang, et al. "Eyes wide shut? exploring the visual shortcomings of multimodal llms." Proceedings of the IEEE/CVF Conference on Computer Vision and Pattern Recognition. 2024.
>
> [2] Wang, Wenxuan, et al. "Diffusion feedback helps clip see better." arXiv preprint arXiv:2407.20171 (2024).
>
> ***
> **Q2&W2**: Ablation experiments on SPL weight
>
> **A2**: Thank you for your valuable feedback. In our manuscript, we conducted a qualitative ablation study to examine the effect of different SPL weight settings, as shown in Fig. 8. Based on these results, we found that with 500 training steps, an SPL weight of 1.0 yielded the best performance since it well follows the text prompt while customizing given dogs. Additionally, we performed a quantitative experiment to further explore the impact of various SPL weight settings. The results of this analysis are presented in the table below. This result shows that CLIP-T becomes higher (increase in the ability to follow prompts) and DINO-I decreases(decrease in the ability to customize concepts) with increasing SPL weight, which is consistent with our expectations. Meanwhile, we find that SPL is a loss function that is insensitive to weight. Combined with the qualitative observation in Fig.8, we prefer to choose 1 as the SPL weight. We will update this quantitative result in the latest version of our manuscript.
>
> | SPL weight | CLIP-T↑ | DINO-I↑ |
> | --- | --- | --- |
> | 0.01 | 0.299 | 0.677 |
> | 0.1 | 0.300 | 0.674 |
> | 1 | 0.300 | 0.673 |
> | 10 | 0.300 | 0.665 |
> | 100 | 0.301 | 0.661 |
>
> ---
>
> **Q3&W3**: Significance of Fig.2
>
> **A3**: Thank you for your insightful feedback. In our manuscript, Figure 1 illustrates the concept of subject-driven personalized generation and highlights the limitations of tuning-based models. In Figure 2, we aim to demonstrate the effectiveness of our model through a story involving a dog and a pair of sunglasses, showing that our model can preserve different subject concepts well， further showcasing a potential real-world application of our approach. Additionally, we acknowledge that some figure captions may have been unclear, potentially leading to misunderstandings. In the latest version of our manuscript, we have revised certain figures and their corresponding captions to enhance clarity.
>
> ---
>
> **Q4**: Why CLIP-T cannot reflect the actual performance
>
> **A4**: Thanks for your valuable feedback. Basically, CLIP-T calculates the similarity between image-text pairs. However, existing works[1, 2] find that CLIP models can hardly tell the fine-grained differences in images, giving a mismatched image-text pair high score consequently. As a result, the CLIP-T score can not fully reflect the performance of the results. We will update this to the latest version of the manuscript to make this section more convincing.
>
> [1] Tong, Shengbang, et al. "Eyes wide shut? exploring the visual shortcomings of multimodal llms." Proceedings of the IEEE/CVF Conference on Computer Vision and Pattern Recognition. 2024.
>
> [2] Wang, Wenxuan, et al. "Diffusion feedback helps clip see better." arXiv preprint arXiv:2407.20171 (2024).

---

> > ### Author Response · Authors · 2024-11-25
> >
> > **Q5**: Clarification on the novelty of BLIP2-T metric
> >
> > **A5**: Though BLIP2-T is not a novel metric proposed by us, we introduced it since the CLIP-T score can hardly reflect the actual performance of personalized models. Some previous works[1-4] have directly demonstrated experimentally that the BLIP2-T score correlates better with human tendencies than the CLIP score in evaluating text-to-image models in the ability to follow text prompts, so we believe it will motivate fairer evaluation of models in this field. We will add a simple description of it to the latest version of our manuscript for clarity. The modification is shown below:
> >
> > > This metric is designed by calculating the similarity between image and text embeddings extracted by the BLIP2 model.
> >
> > [1] Grimal, Paul, et al. "TIAM-A metric for evaluating alignment in Text-to-Image generation." Proceedings of the IEEE/CVF Winter Conference on Applications of Computer Vision. 2024.
> >
> > [2] Singh, Jaskirat, and Liang Zheng. "Divide, evaluate, and refine: Evaluating and improving text-to-image alignment with iterative vqa feedback." Advances in Neural Information Processing Systems 36 (2023): 70799-70811.
> >
> > [3] Wu, Xiaoshi, et al. "Human preference score v2: A solid benchmark for evaluating human preferences of text-to-image synthesis." arXiv preprint arXiv:2306.09341 (2023).
> >
> > [4] Xu, Jiazheng, et al. "Imagereward: Learning and evaluating human preferences for text-to-image generation." Advances in Neural Information Processing Systems 36 (2024).
> >
> > ---
> >
> > **Q6**: Explanation of Fig.3a
> >
> > **A6**: Fig.3a is an overview of our first experimental result. It shows that after fine-tuning, the semantic distance (Geometric distance of phrases containing special characters (e.g. \<sks\> dog) and phrases containing only category centers (e.g. dog) after dimensionality reduction of CLIP space to 2D). This indicates that the semantics of the personalized concepts change during fine-tuning. In short, the model increasingly fails to recognize that personalized concepts belong to the dog category.

---

> > > ### Author Response · Authors · 2024-11-30
> > > **Look forward to your feedback**
> > >
> > > Dear Reviewer BPUN,
> > >
> > > Thank you for your thorough review and thoughtful assessment of our work! Since there are only few days remaining for discussion, we look forward to your feedback on our responses.
> > >
> > > Should there be any remaining concerns or areas that require further clarification, we are happy to address them.
> > >
> > > Best regards,
> > >
> > > ClassDiffusion Authors

---

> > > > ### Comment · Reviewer_BPUN · 2024-12-02
> > > > **Rely to authors**
> > > >
> > > > I commend the authors for their efforts in addressing some of my concerns in the revised manuscript. Although the changes have partially resolved specific issues highlighted in my initial review, I have carefully assessed the manuscript's overall quality alongside the feedback from other reviewers. After thorough consideration, I have decided to improve the score to 5.

---

> > > > > ### Author Response · Authors · 2024-12-02
> > > > >
> > > > > Dear Reviewer BPUN,
> > > > >
> > > > > We appreciate you for your review. We are glad to see that our reply addressed some of your concerns.
> > > > >
> > > > > Best regards,
> > > > >
> > > > > ClassDiffusion Authors

---

### Official Review · Reviewer_FpQ7 · 2024-11-03

**Soundness:** 3
**Presentation:** 3
**Contribution:** 3
**Rating:** 6
**Confidence:** 2

**Summary:**

This paper addresses a critical issue in text-to-image personalization models: the loss of compositional ability after fine-tuning. The authors identify that after personalization tuning, models struggle to generate images that combine the personalized concept with other elements (e.g., a personalized dog wearing headphones). Through experimental and theoretical analysis, they attribute this to semantic drift of the personalized concept from its superclass. They propose ClassDiffusion, which introduces a semantic preservation loss to maintain alignment between personalized concepts and their superclass during fine-tuning.

**Strengths:**

1 The paper provides a thorough investigation of a significant problem in personalization models, supported by both empirical observations and theoretical analysis.

2 The proposed solution is elegantly simple yet effective, making it practical for implementation.

3 The experimental results demonstrate clear improvements over existing methods.

**Weaknesses:**

1 The paper lacks comprehensive ablation studies to demonstrate the impact of different components of the semantic preservation loss.

2 While the authors mention extending their method to video generation, this aspect feels somewhat underdeveloped and could benefit from more detailed exploration.

**Questions:**

1 Include more detailed ablation studies to validate the design choices in the semantic preservation loss.

2 Expand the evaluation section with more quantitative metrics specifically designed to measure compositional ability.

3 Provide more extensive comparisons with other state-of-the-art methods that attempt to address similar issues. Include more diverse case studies beyond the dog-headphone example to demonstrate the generality of the approach.

---

> ### Author Response · Authors · 2024-11-25
>
> **Q1&W1**: Ablation on components of SPL
>
> **A1**: We have a qualitative ablation on the weight of semantic preservation loss(SPL) in our manuscript. Also, we agree with your assertion that a more comprehensive quantitative ablation experiment on its weight may be beneficial for demonstrating the impact of SPL, which shows Experiment results are shown below. The results demonstrate that as the SPL weight increases, CLIP-T scores rise (indicating better prompt-following ability) while DINO-I scores decline (showing reduced concept customization ability). This trend aligns with our expectations. Notably, we observe that the SPL loss function exhibits low sensitivity to weight variations. Based on these findings and the qualitative results shown in Fig.8, we selected 1 as the optimal SPL weight. we update this result in the latest version of our manuscript.
>
> | SPL weight | CLIP-T↑ | DINO-I↑ |
> | --- | --- | --- |
> | 0.01 | 0.299 | 0.677 |
> | 0.1 | 0.300 | 0.674 |
> | 1 | 0.300 | 0.673 |
> | 10 | 0.300 | 0.6652 |
> | 100 | 0.301 | 0.6614 |
>
> ---
>
> **Q2**: More quantitative experiments
>
> **A2**: We believe adapting some quantitative metrics beyond CLIP-T, CLIP-I, DINO-I, and BLIP2-T is an instructive and interesting idea though it is not so common in the field of subject-driven personalized generation. We adopt a widely used method called TIFA[1] to evaluate the compositional ability of models.
>
> Experiment results are shown below, we update this result to the latest version of our manuscript.
>
> | Models | TIFA-score↑ |
> | --- | --- |
> | DreamBooth | 0.559 |
> | Textual Inversion | 0.505 |
> | Custom Diffusion | 0.746 |
> | NeTI | 0.607 |
> | SvDiff | 0.835 |
> | Our | 0.843 |
>
> [1] Hu, Yushi, et al. "Tifa: Accurate and interpretable text-to-image faithfulness evaluation with question answering." *Proceedings of the IEEE/CVF International Conference on Computer Vision*. 2023.
>
> ---
>
> **Q3**: More comparison with additional baseline that focus on similar problems
>
> **A3**: We have added Prospect[1], a state-of-the-art baseline for subject-driven personalized generation, to demonstrate our model's effectiveness. The quantitative results below show that our model achieves superior performance. We have included these results in our manuscript's appendix. For diverse cases, Fig.2, Fig.6, Fig.7 and Fig.13 show some qualitative results beyond the dog-headphone example, showing the effectiveness of our proposed method.
>
> | Model | CLIP-T↑ | CLIP-I↑ | DINO-I↑ |
> | --- | --- | --- | --- |
> | DreamBooth | 0.249 | 0.855 | 0.700 |
> | Textual Inversion | 0.242 | 0.825 | 0.631 |
> | Custom Diffusion | 0.286 | 0.837 | 0.693 |
> | NeTI | 0.290 | 0.838 | 0.648 |
> | SvDiff | 0.293 | 0.834 | 0.606 |
> | Prospect[1] | 0.294 | 0.815 | 0.588 |
> | ClassDiffusion(Our) | 0.300 | 0.828 | 0.673 |
>
> [1] Zhang, Yuxin, et al. "Prospect: Prompt spectrum for attribute-aware personalization of diffusion models." *ACM Transactions on Graphics (TOG)* 42.6 (2023): 1-14.
>
> ---
>
> **W2**: Exploration in video generation
>
> **A4**: Thank you for your comments. We conducted an experiment on subject-driven personalized video generation to demonstrate that ClassDiffusion can generate personalized videos while preserving the semantic space—without requiring additional computational resources for fine-tuning. Also, we believe that for almost all methods in video generation that obtain video generation ability by combining 2D stable diffusion weights with time adapters, our method achieves substantial results in video generation without the need for extra fine-tuning computations. In this way, we can use 2D fine-tuned parameters in image-driven video editing or some other tasks.

---

> > ### Author Response · Authors · 2024-11-30
> >
> > Dear Reviewer FpQ7,
> >
> > Thank you for your thorough review and thoughtful assessment of our work! Since there are only few days remaining for discussion, we look forward to your feedback on our responses.
> >
> > Should there be any remaining concerns or areas that require further clarification, we are happy to address them.
> >
> > Best regards,
> >
> > ClassDiffusion Authors

---

> > > ### Comment · Reviewer_FpQ7 · 2024-12-02
> > >
> > > I appreciate the authors' efforts in addressing some of my concerns in their revision. While the changes have partially resolved certain issues I raised in my initial review, I have carefully considered the manuscript's overall quality and other reviewers' comments. After thorough evaluation, I maintain my original score of 6 points.

---

> > > > ### Author Response · Authors · 2024-12-02
> > > >
> > > > Dear Reviewer FpQ7,
> > > >
> > > > Thank you for your response, we are glad to see that our response addressed some of your concerns.
> > > >
> > > > Best regards,
> > > >
> > > > ClassDiffusion Authors

---

### Official Review · Reviewer_Jqzc · 2024-11-03

**Soundness:** 3
**Presentation:** 3
**Contribution:** 3
**Rating:** 6
**Confidence:** 4

**Summary:**

The manuscript introduces "ClassDiffusion," a methodology for personalizing text-to-image generation. Leveraging a semantic preservation loss, this method facilitates rapid and efficient concept customization, and prevent the semantic drift during the fine-tuning process. The results demonstrate that the proposed work is competitive with leading frameworks in various image generation tasks.

**Strengths:**

1. The proposed work observes the reason why the existing methods suffer from the issue of compositional ability with experiments and theoretical analysis.

2. The authors have developed a framework for generating personalized images that effectively ensure the compositional capabilities for concept personalization tuning.

2. The paper provides experimental results, including both quantitative and qualitative assessments, showcasing the superior performance of the framework. The results clearly highlight the effectiveness of the proposed method in facilitating personalized image generation.

**Weaknesses:**

1. In Figure 5, the method utilizes the semantic preservation loss. However, it is unclear how it would perform with fine-grained subjects (two dogs or two cats with different breed). Clarification on whether the method can effectively manage such fine distinctions and a visual representation of joint embeddings for both similar and diverse subjects would be beneficial.

2. Recent methodologies [1, 2] have demonstrated the capability to learn multi-concept personalization, it remains uncertain if the proposed work can handle multiple personalized instances, particularly for contexts involving up to five subjects. Although quantitative results for two subjects are provided in the paper, the absence of qualitative results for three or more subjects in both the main text and appendix is a notable omission. Including these results would substantiate the method's capability in more complex scenarios.

[1] Liu, Zhiheng, et al. "Cones 2: Customizable image synthesis with multiple subjects." arXiv preprint arXiv:2305.19327 (2023).

[2] Yeh, Chun-Hsiao, et al. "Gen4Gen: Generative Data Pipeline for Generative Multi-Concept Composition." arXiv preprint arXiv:2402.15504 (2024).

**Questions:**

Given the concerns mentioned, particularly around the method's scalability to more complex multi-subject personalizations and the rationale behind the loss choices, I recommend a "marginally below the acceptance threshold" for this paper. Enhancements in demonstrating multi-subject capabilities, clarity in embedding visualization, and justification for the choice of technology could potentially elevate the manuscript to meet publication standards.

---

> ### Author Response · Authors · 2024-11-25
>
> **Q1-1**: Rationale behind the loss choice
>
> **A1-1**: We refine the explanation of our proposed method in $\S$3.4 to explain the rationale behind the loss choice more clearly. It‘s mainly because according to our experimental observation, one perspective to solve this problem lies in preserving semantic information during fine-tuning to reduce the difficulty of sampling from the joint conditional distribution. To address this, we propose a novel loss function aimed at constraining semantic variation throughout the fine-tuning process. So we design a loss function to reduce the change in the semantic distance between class prompt(*e.g.* a photo of a dog) and training prompt(*e.g.* a photo of a V* dog).
>
> ---
>
> **Q1-2**: Clarity in embedding visualization
>
> **A1-2**: Thanks for your valuable feedback. We refine our manuscript by adding the following details about embedding visualization. The modification is shown below.
> > First, we collect 70 phrases that combine adjectives with words representing the class of given concepts (e.g., "a happy dog" or "a cool dog"). Using the CLIP text encoder, we extract text embeddings for these phrases. To visualize the semantic space and intuitively track modifications within it, we use t-SNE to reduce these high-dimensional embedding vectors to 2D. An overview of this result are shown in Fig.3(a), meanwhile we show the real experimental results in Fig.12.
>
> ---
>
> **W1**: Different Fine-grained concepts
>
> **A1**: Thank you for your thoughtful remarks. We completely agree that developing methods for addressing fine-grained customization is an important direction of research. However, we would like to clarify that our proposed method is not specifically designed to address fine-grained customization problems.
>
> Besides, they are two isolated problems since we found the problem of reduced combinatorial power to the increased difficulty of sampling the joint probability distribution of the two classes, yet the fine-grained generation problem is an intraclass probability distribution problem.
>
> Therefore, we believe that fine-grained customization and the reduction of compositional abilities after personalized fine-tuning represent distinct challenges that require fundamentally different approaches. In our work, we focus on addressing the reduction in compositional abilities observed after the personalized fine-tuning process. Through our experimental and theoretical analyses, we identified the primary issue as the increased difficulty in sampling from the joint conditional distribution. This observation motivated us to propose the Semantic Preservation Loss (SPL), which directly addresses this challenge. We hope this explanation helps to clarify our focus and the contributions of our work.
>
> ---
>
> **W2**: Multiple concepts ($\ge$ 3)
>
> **A2**: We conducted quantitative experiments on customizing multiple concepts and added the results to our manuscript's appendix. The experiments demonstrate that ClassDiffusion generates high-quality results when combining multiple concepts, validating the effectiveness of our proposed methods. Please refer to Figure 13 for the experiment result.

---

> > ### Comment · Reviewer_Jqzc · 2024-11-27
> > **Response to Author's Rebuttal**
> >
> > The rebuttal addressed most of my questions and concerns, and I appreciate the clear and thorough explanation. After reviewing the authors' response carefully, I still have reservations about the multiple concept generation aspect. Specifically, based on the "single example" provided for each composition in Figure 13, the model seems to face challenges in generating multiple concepts beyond two. Have the authors elaborated on the reasons behind this limitation?
> >
> > That said, since most of my concerns have been addressed, I am raising my score to 6.

---

> > > ### Author Response · Authors · 2024-11-30
> > >
> > > Dear Reviewer Jqzc.
> > >
> > > Thank you for your valuable feedback and kind acknowledgment. We are glad to know that our rebuttal addresses most of your questions and concerns.
> > >
> > > For the multiple concept generation, we lean towards it’s a good result from the perspective of the ability to follow text prompts(in both cases, our model succeeds in generating all concepts.) However, we recognize that from the perspective of customizing the given concepts, it is somehow not so excellent as it misses some fine-grained features of given features including the material of the chair back and slight color shift in sun-glasses.
> > >
> > > We think there is a potential reason: The inherent ability of Stable Diffusion v1.5.  Compared to some existing SOTA text-to-image models(*e.g.* FLUX, Stable Diffusion 3.5), Stable Diffusion v1.5 lacks the ability to generate multiple subjects in one image without additional guidance[1]. This leads to the natural difficulty of fine-tuning the customization of multiple concepts without additional guidance.
> > >
> > > Overall, customizing multiple concepts without additional guidance is a challenging open problem that is worth further exploration in my opinion. We agree with your emphasis on this open problem, and we look forward to further work in this direction.
> > >
> > > Thank you again for your valuable feedback and insightful ideas.
> > >
> > >
> > > Best Regards,
> > >
> > > ClassDiffusion Authors.
> > >
> > > [1] Chefer, Hila, et al. "Attend-and-excite: Attention-based semantic guidance for text-to-image diffusion models." *ACM Transactions on Graphics (TOG)* 42.4 (2023): 1-10.

---

### Official Review · Reviewer_pV4W · 2024-11-04

**Soundness:** 3
**Presentation:** 3
**Contribution:** 2
**Rating:** 8
**Confidence:** 4

**Summary:**

This work delves into the captivating realm of text-to-image customization. However, current tuning-based methods often struggle with a common pitfall: they tend to overfit to the limited concepts presented in their training datasets. This overfitting can significantly diminish the model's compositional abilities. The authors highlight that this decline is mainly due to the semantic shifts that occur with customized concepts during the fine-tuning process.  To address this issue, the authors introduce an approach called ClassDiffusion. This technique revitalizes the model's compositional strength by restoring its connection to the original semantic space. Through a series of thorough experiments, the authors demonstrate the effectiveness of ClassDiffusion and the fresh insights it brings to interconnected fields. The results not only showcase its prowess but also pave the way for exciting advancements in the domain.

**Strengths:**

1. The authors unveil the intriguing underlying cause behind the overfitting of SD. Rather than a typical case of overfitting, they suggest it’s more about semantic drift—a perspective I find quite compelling, though I still believe overfitting truly occurs.

2. This paper is not only well-written but also flows smoothly, making it a pleasure to read.

3. The inclusion of additional relevant baselines demonstrates the authors' thorough research and deep engagement with the topic.

4. The authors provide a wealth of comprehensive experimental results, reinforcing the strength of their proposed ideas.

**Weaknesses:**

1. I agree that semantic drift occurs during fine-tuning. This drift arises because the model is an intricate network filled with numerous parameters. When we fine-tune with only few samples, the adjustments to these parameters are minimal, yet the model still retains its generative capabilities. In my view, semantic loss may struggle to restore the original semantic space, as it doesn’t preserve the nuances of the drift encountered. Theoretically, utilizing the entire training text data could mitigate this issue.

2. While I think that the proposed method is quite straightforward and effective,  it should be explained further.

3. The figures in this paper could use some enhancement—except for Figure 4, which stands out nicely. The other figures are somewhat simplistic, and certain visuals, like Figure 2, lack clarity when paired with their captions.

4. Although the writing is commendable, both the tables and figures require refinement. For instance, in Table 1, every similarity is bolded, which could lead to confusion.

**Questions:**

see weaknesses

---

> ### Author Response · Authors · 2024-11-25
>
> **W1**: SPL may struggle to restore the original semantic space
>
> **A1**: I completely agree with your perspective that semantic preservation loss (SPL) cannot fully restore the original semantic space.  In fact, I believe that even minor alterations in the semantic space are exceedingly challenging to fully restore due to their occurrence within a high-dimensional space.  Consequently, restoring the original semantic space solely through simple regularization is inherently difficult. Moreover, a completely unchanged semantic space may hinder the customization of specific concepts, since we need to fine-tune the embedding to adapt for given concepts. Our SPL is designed to preserve semantic space in terms of minimizing the semantic distance between the class prompt(*e.g. a photo of a dog*) and the training prompt(*e.g. a photo of a V\* dog*). In this way, though changes still happen in semantic space which might be inevitable, we are successful in keeping the model's ability of following prompts.
>
> ***
>
> **W2**: Further explanation of the proposed method
>
> **A2**: We construct the exposition of our proposed method from high level (the design concept of our proposed method) to low-level technical details (the computational method of our proposed method to the formalized formula) to express our method more clearly.
>
> According to our experimental observation and theoretical analysis, The key problem is that it is more difficult to sample from the joint probability distribution of the two classes due to semantic drift. Therefore, the core challenge addressed is preserving semantic information during fine-tuning to simplify sampling from the joint conditional distribution. To tackle this, a novel semantic preservation loss (SPL) is proposed. This loss minimizes the semantic distance between the target concept phrase (e.g., "a photo of a V* dog") and the general class phrase (*e.g.*, "a photo of a dog"). Using Stable Diffusion’s text encoder, embeddings for these phrases are computed, and their cosine distance is used to calculate SPL. The final loss function combines SPL with the reconstruction loss, weighted by a factor $\lambda$, ensuring the fine-tuning process aligns customized token embeddings with their respective target concepts while maintaining semantic consistency.
>
> Also out of consideration for clarity, we separate method design and implementation details into different sections. Modifications are conducted in Sections 3.4 and 4.1, please refer to our latest manuscript for these modifications.
>
> ***
> **Q3**: Caption & Clarification on Figure
>
> **A3**: We refine the caption of Figure 2 in the latest version of our manuscript. Figure 2 presents An example of a complete narrative. For example, the above showcases a bear’s literary journey: from reading a book to ultimately earning a Nobel Literature Prize. This example highlights a potential real-world application enabled since our model can combine customized concepts with other elements. Modification is shown below
>
> > Figure.2 A qualitative result of two small stories produced by our model. The above showcases a bear's literary journey: from reading a book to ultimately earning a Nobel Literature Prize. The below shows the fate of sunglasses. Finally, the bear gets the sunglasses. It shows a potential real-world application due to our model's high performance.
>
> ***
> **W4**: Refinement on figures and tables.
>
> **A4**: We totally agree that refinements on figures and tables in the manuscript will be beneficial for clarity. As we stated above, we found some figures and captions under the consideration of clarity. Besides, we will refine the presentation of the user study's result in Table 1 and the corresponding caption in the latest version of our manuscript.

---

> ### Comment · Reviewer_pV4W · 2024-11-26
> **Official Comment by Reviewer pV4W**
>
> Many thanks to the authors for their detailed response, which has addressed most of my concerns. This paper proposes an efficient method to overcome the overfitting, resulting in failure to create the concept under multiple conditions. Authors perform an in-depth study of the properties and applications of the base model. In this paper, a more important observation is that the compositional ability only disappears after personalization tuning. Equipped with this better understanding, this paper leverages these properties to solve a diverse set of tasks. I raise my score.

---

> ### Author Response · Authors · 2024-11-26
>
> Dear Reviewer pV4W,
>
> Thank you for your kind acknowledgment. We are delighted to know that all concerns have been addressed to your satisfaction.
>
> Best regards,
> The Authors

---

### Meta-Review · Area_Chair_8TP3 · 2024-12-22

**Metareview:**

The manuscript received ratings of 6, 6, 8, and 5. Reviewers appreciated that the manuscript studies the underlying cause behing overfitting of Stable Diffusion and the reasons behing the struggle of existing works with respect to the issue of compositional ability. Reviewers also rasied some concerns in the initial review, including additional details about the utilization of the semantic preservation loss, effectiveness of the proposed method to handle multiple personalized instances, additional comparisons using the quantitative metrics specifically designed to measure compositional ability, and more clarity with respect to figure 3a. Authors provided a rebuttal to address the concerns of the reviewers, including further explanation of the proposed method (e.g., rationale behind the loss choice), improving caption & clarification about the figure, more details about embedding visualization, and quantitative experiments on customizing multiple concepts. Post-rebuttal, three reviewers remained positive mentioning that most of their concerns are addressed in the rebuttal.  Given the reviewers comments, rebuttal and discussions, the recommendation is accept. Authors are strongly encouraged to take into consideration reviewers feedback when preparing the revised manuscript.

**Additional Comments On Reviewer Discussion:**

Reviewers appreciated that the manuscript studies the underlying cause behing overfitting of Stable Diffusion and the reasons behing the struggle of existing works with respect to the issue of compositional ability. Reviewers also rasied some concerns in the initial review, including additional details about the utilization of the semantic preservation loss, effectiveness of the proposed method to handle multiple personalized instances, additional comparisons using the quantitative metrics specifically designed to measure compositional ability, and more clarity with respect to figure 3a. Authors provided a rebuttal to address the concerns of the reviewers, including further explanation of the proposed method (e.g., rationale behind the loss choice), improving caption & clarification about the figure, more details about embedding visualization, and quantitative experiments on customizing multiple concepts. Post-rebuttal, three reviewers remained positive mentioning that most of their concerns are addressed in the rebuttal.

---

### Decision · Program_Chairs · 2025-01-22

Accept (Poster)